# Feasibility assessment of double-blind, crossover, randomized controlled trial protocol comparing two oxygen-supplemented pulmonary rehabilitation for patients with chronic obstructive pulmonary disease: A pilot study

Akihiro Ito[1]*[⊙], Akane Morito[2], Masahiro Ishizaka[1], Yukihiro Ogawa[2], Yuki Kawai[2], Yuta Hanawa[2], Naotaka Onodera[2], Yoshiaki Endo[1], Isato Fukushi[3], Kotaro Takeda[4], Taichi Mochizuki[5], Yasushi Inoue[5], Yasuo To[6], Seiichiro Sakao[6], Kazuyuki Chibana[7], Hideaki Yamasawa[8], Satoshi Fuke[9], Sarah Kesler[10], David Gozal[11], Yasumasa Okada[12], Akira Umeda[13]*[⊙]

1 Department of Physical Therapy, School of Health Science, International University of Health and Welfare, Ohtawara-City, Japan, 2 Department of Rehabilitation, International University of Health and Welfare Shioya Hospital, Yaita-City, Japan, 3 Graduate School of Health Sciences, Aomori University of Health and Welfare, Aomori-City, Japan, 4 Faculty of Rehabilitation, School of Health Sciences, Fujita Health University, Toyoake-City, Japan, 5 Department of Respiratory Diseases Center, International University of Health and Welfare Mita Hospital, Tokyo, Japan, 6 Department of General Medicine, St. Marianna University School of Medicine, Kawasaki-City, Japan, 7 Department of Pulmonary Medicine, Dokkyo Medical University, Nikko Medical Center, Nikko-City, Japan, 8 Department of Pulmonary Medicine, International University of Health and Welfare Hospital, Nasushiobara-City, Japan, 9 Department of Respiratory Diseases Center, KKR Sapporo Medical Center, Sapporo-City, Japan, 10 Intensive Care Unit, University of Minnesota, Minneapolis, Minnesota, United States of America, 11 Joan C. Edwards School of Medicine, Marshall University, Huntington, West Virginia, United States of America, 12 Department of Internal Medicine, Murayama Medical Center, Musashimurayama-City, Tokyo, Japan, 13 Department of General Medicine, School of Medicine, IUHW Shioya Hospital, International University of Health and Welfare (IUHW), Yaita-City, Japan

⊙ These authors contributed equally to this work.
* umeda@ihwg.jp (AU); i.akihiro@ihwg.ac.jp (AI)

## Abstract

### Background

Pulmonary rehabilitation (PR) for patients with chronic obstructive pulmonary disease (COPD) improves exercise tolerance and COPD assessment test score (CAT). Oxygen supplementation during PR facilitates exercise physiological benefits. This study aimed to assess the feasibility of a trial comparing two oxygen supplementation methods, with the hypothesis that both would be effective and produce distinct outcomes.

### Methods

This double-blind, crossover, randomized controlled trial compared two PR programs—Program A (including PR under $FiO_2$ 0.3) and Program B (including PR under $FiO_2$ 0.5)—using high-flow nasal cannula oxygen therapy in patients with COPD

**Data availability statement:** All relevant data are within the paper and its Supporting Information files.

**Funding:** AI 24K20446 Grants-in-Aid for Scientific Research（KAKENHI）https://www.jsps.go.jp/j-grantsinaid/ co-first author, co-corresponding author.

**Competing interests:** NO authors have competing interests.

and exertional dyspnea (n = 6). Data on the 6-minute walk distance (6MWD), CAT, muscle strength, body composition analysis, respiratory function, and joint range of motion were collected. Participants underwent one month of regular PR followed by two months of oxygen-supplemented PR, with data collected again after this period. Statistical significance was set at 0.05 with a power of 0.8, and the required sample size was calculated accordingly.

## Results

The required sample size could not be calculated based on the 6MWD. The improvement in CAT by Program A was greater than that by Program B. The improvements in muscle parameters by Program B were greater than those by Program A. The standardized effect size and the corresponding required sample sizes for the CAT, quadriceps muscle power, lower leg circumference, trunk muscle mass, and leg muscle mass were 0.32/81, 0.66/8, 0.17/114, 0.27/88, and 0.24/56, respectively.

## Conclusions

Given the small number of participants, the 6MWD and CAT were not appropriate primary endpoints for comparing the effectiveness of the two oxygen supplementations during PR in patients with COPD. However, the quadriceps muscle power was identified as the most suitable primary endpoint among all the investigated parameters.

## Introduction

Chronic obstructive pulmonary disease (COPD) is among the top three causes of deaths worldwide, alongside cardiovascular diseases and cancer, with nearly 90% of COPD-related deaths occuring in low- and middle-income countries [1]. The prevalence of COPD varies widely by region, age, and the availability of diagnostic spirometry [1]. It accounts for approximately 5% of respiratory-related deaths and is a leading cause of death in the older population [1]. COPD is caused by smoking, air pollution, biomass exposure, occupational dusts, and host factors (including abnormal lung development and lung aging) and is characterized by reduced alveolar ventilation due to alveolar destruction and ventilation–blood flow imbalance [1,2]. This results in dyspnea, chronic cough, and increased sputum production [3]. Additionally, it causes a decline in physical function, such as exercise tolerance and muscle weakness, and reduces quality of life [4,5].

Pulmonary rehabilitation (PR) plays an important role in the management of patients with COPD by improving exercise tolerance and quality of life and reducing dyspnea and fatigue [6–12]. The PR program for patients with stable COPD typically comprises supervised exercise training lasting approximately 4–12 weeks, generally combining strength training with aerobic exercise [13–15]. Oxygen therapy combined with exercise for patients with COPD has been increasingly evaluated and adopted [16–19]. Evidence is gradually emerging regarding the intervention effects of PR combined with oxygen therapy [20].

Recently, there has been growing interest in the potential benefits of oxygen therapy with a high-flow nasal cannula (HFNC) for COPD. HFNC reduces respiratory muscle load by lowering the arterial partial pressure of carbon dioxide, increasing end-expiratory and end-tidal volumes, and decreasing the respiratory rate, which may improve respiratory patterns [21–23]. Rehabilitation with HFNC has been shown to be superior to conventional rehabilitation using a nasal cannula or Venturi mask in improving the 6-minute walk distance (6MWD) and exercise endurance time in patients with COPD [24,25]. Additionally, exercise training with HFNC may be superior to exercise training with a regular nasal cannula in patients with chronic respiratory failure receiving long-term oxygen therapy [26].

However, most of these studies have focused on *in-situ* observations with few interventional studies. Additionally, the optimal setting for HFNC has not been adequately investigated. A meta-analysis of HFNC intervention effects found that while some improvements in quality of life and exercise tolerance were observed, the evidence was insufficient [27]. This lack of consistency was partly due to varying HFNC settings across studies.

Therefore, this study aimed to assess the feasibility of a trial comparing HFNC at two $FiO_2$ levels, with the hypothesis that both methods would be effective but differ in their effects. This study may maximize the effects of PR, further improve patients' exercise capacity and quality of life, and reduce dyspnea. Moreover, identifying the optimal HFNC setting could enhance its clinical utility for patients with COPD.

## Materials and methods

### Setting and participants

This double-blind, crossover, randomized controlled trial was reviewed and approved by the International University of Health and Welfare Ethics Committee according to the Declaration of Helsinki (approval number: 21-B-4). All patients provided written informed consent to participate in the study. This study is registered at the University Hospital Medical Information Network (UMIN) Clinical Trials Registry (registration number: UMIN000047507). The protocol described in this peer-reviewed article is published on protocols.io (dx.doi.org/10.17504/protocols.io.x54v9bdxml3e/v1) and is included for printing purposes as S1 File.

Patients with COPD with dyspnea on exertion were recruited from the International University of Health and Welfare (IUHW) Shioya Hospital. The diagnosis of COPD was based on forced spirometry showing the presence of a post-bronchodilator forced expiratory volume in 1 s (FEV1)/ forced vital capacity (FVC) < 0.7 [1,28]. Six participants were consecutively recruited for the pilot study from February 2022 to August 2022. At least nine months per participant were required to record the full data.

### Protocol of a double-blind, crossover, randomized controlled trial comparing two oxygen-supplemented PR programs

Study participants were randomly assigned to one of the two groups using random number tables: one group started a PR program with oxygen supplementation at $FiO_2$ 3 (Program A), and the other began a PR program with oxygen supplementation at $FiO_2$ 0.5 (Program B) (Fig 1). The protocol developed in this study was based on the measurement methods employed in prior research [29,30]. These assignments were concealed from both the participants and the physical therapists responsible for the PR program.

The study protocol is illustrated in Fig 1. The participants underwent regular rehabilitation without oxygen supplementation once a week for four sessions (approximately 1 month). They then underwent eight PR sessions once a week (approximately 2 months) with assigned oxygen supplementation. After the completion of the intervention, a washout period of > 3 months was established. Following the washout period, PR was performed once a week for 12 sessions (approximately 2 months) with crossover oxygen supplementation. The washout period was set according to prior

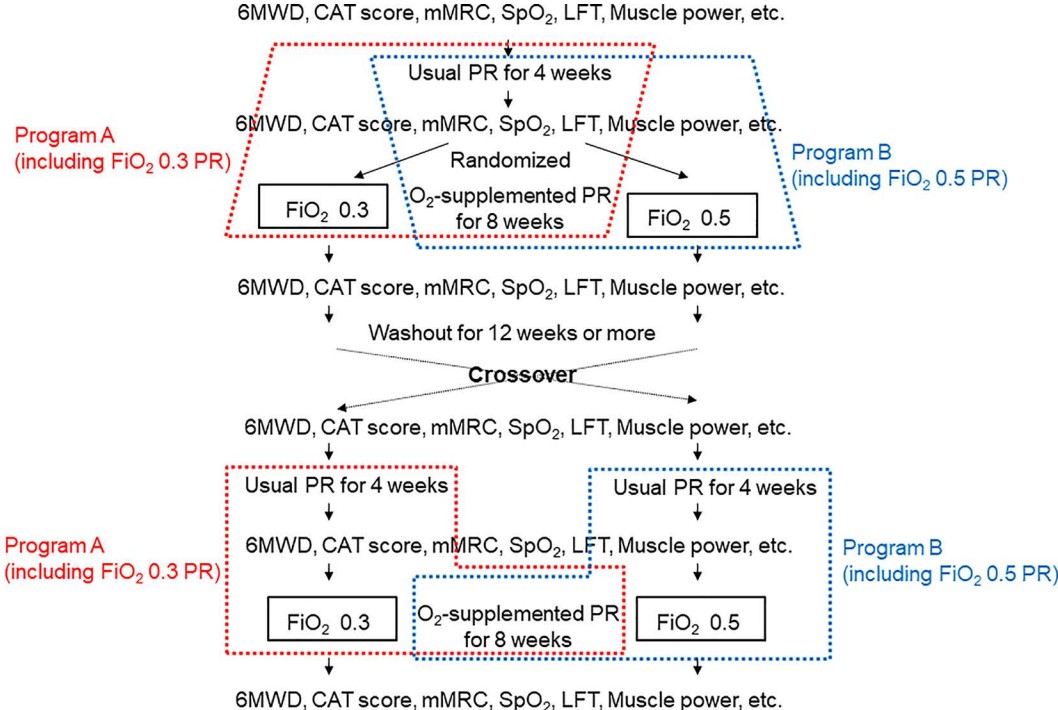

**Fig 1. Protocol.** A double-blind, crossover, randomized controlled trial comparing Program A, including 4 weeks of usual PR followed by 8 weeks of PR under FiO₂ 0.3, with Program B, including 4 weeks of usual PR followed by 8 weeks of PR under FiO₂ 0.5. The washout period was set for 12 weeks or more. Abbreviations: 6MWD, Six-Minute Walk Distance; CAT, COPD Assessment Test; mMRC, Modified Medical Research Council Dyspnea Scale; LFT, Lung function tests; PR, pulmonary rehabilitation.

research to allow sufficient time for the learning effects of exercise therapy and the intervention effects of aerobic exercise to dissipate [31–33].

A total of six assessments were conducted: at baseline; after the first course of standard rehabilitation; after the first course of O₂-supplemented PR; after the washout period; after the second course of standard rehabilitation; and at the end of the second course of O₂-supplemented PR. Evaluations included 6MWD, COPD assessment test (CAT), modified Medical Research Council (mMRC) dyspnea scale, quadriceps strength at knee extension, lower leg circumference, and body composition. Respiratory function was assessed on four occasions: at baseline, after the first course of PR, after the washout period, and at the end of the second course of PR.

The 6MWD was measured once, following the protocol from previous studies [34,35]; however, the distance for a walking round trip was set to 15 m in this study. Participants walked at their own pace in comfortable clothing, aiming to cover as much distance as possible in 6 min. Rest breaks were allowed during walking, and the physiotherapist in-charge encouraged participants to rest if subjective fatigue, dyspnea, or a significant drop in SpO₂ (below 90%) were observed. Participants were instructed to walk slowly if SpO₂ fell below 85% and to stop if necessary. Blood pressure, respiratory rate, heart rate, dyspnea, and lower limb fatigue were measured before and after the measurements using the Borg scale.

CAT and mMRC scores were obtained directly from participants using specialized measurement forms [36,37]. Lower scores indicate better conditions for both scores.

The knee extensor strength was measured using a Mobie dynamometer (SAKAI med, Japan). The participant was seated with the knee joint flexed at 90°, and a band was attached to the distal leg. Isometric contraction was used to assess bilateral muscle strength [38]. Measurements were taken twice, and the maximum value was used.

The circumference of the lower leg was measured using a tape at the point of maximum circumference. Measurements were taken three times, and the median value was used [39].

Body composition was measured using a MC-780A-N body composition analyzer (TANITA, Japan). The participants were assessed in a standing position with their feet bare. The measurement periods were set to be the same. Limb and trunk site-specific total muscle mass, body fat percentage, body fat mass, and lean body mass were measured.

Respiratory function was measured using a CHESTAC-8900 respiratory function tester (CHEST, Tokyo, Japan). Lung capacity and forced vital capacity tests were performed using standardized methods [40,41]. The reference values for the FEV1 predicted values were derived from the Japanese standards [42].

The range of motion (trunk flexion and extension) was measured using the methods of the Japanese Orthopaedic Association, Japanese Association of Rehabilitation Medicine, and Japanese Society of Foot Surgery. Measurements were taken from the lateral aspect of the trunk, with the posterior sacrum as the base axis and the line connecting the first thoracic spinous process and the fifth lumbar spinous process as the axis of motion. Care was taken to ensure that hip motion was not included.

## Pulmonary rehabilitation under $O_2$ supplementation with HFNC

The $O_2$-supplemented PR protocol consisted of conditioning, resistance exercise (approximately 20 min), resting on a chair, and walking on a treadmill for 20 min under $O_2$ supplementation with HFNC. PR was conducted for approximately 40 min once a week [43,44]. Conditioning focused on the relaxation and stretching of the thoracic and respiratory muscles. Resistance exercises were performed mainly on the lower extremities while monitoring for dyspnea. The treadmill speed was adjusted to target a breathing difficulty level of 4–5 on the revised Borg scale, which is a guideline for aerobic exercise, and the discontinuation criteria were based on the guidelines of the American College of Sports Medicine [45]. Oxygen supplementation was provided using an NKV-330 ventilator (Nihon Kohden), with a flow rate set at 20 L/m unless otherwise indicated. The $FiO_2$ was set at 0.3 or 0.5. HFNC was initiated just prior to the start of conditioning and discontinued after the end of the walking exercise on a treadmill, when the participant was resting in a sitting position [29].

During the rehabilitation intervention, the Transcutaneous Monitor-4 (Radiometer Medical ApS, Denmark) was used to monitor $PtcCO_2$ and $PtcO_2$. Since $PtcCO_2$ values are typically higher than $PaCO_2$ values, many commercially available instruments have adopted a post-measurement adjustment approach, such as subtracting 4–5 mmHg from the directly measured $PtcCO_2$ values for better estimation of the adult $PaCO_2$. However, we did not adopt such an adjustment; both $PtcCO_2$ and $PtcO_2$ were recorded directly from the tcSensor 84 sensor (Radiometer Medical ApS, Denmark). This approach was applied consistently across all subject runs [29,46,47]. Skin sensors were applied to the forearms of the participants following the manufacturer's recommendations.

## Safety of HFNC settings

Previous studies on respiratory rehabilitation interventions using HFNC have employed flow rates ranging from 20 to 60 L/min, which is inconsistent [27]. Concerns have been raised that both low and high flow rates could lead to issues, such as $CO_2$ accumulation and over-reduction. Additionally, high flow rates may increase discomfort and should be considered [48]. Based on the results of previous studies, $PtcCO_2$ and $PtcO_2$ were monitored during rehabilitation using the Transcutaneous Monitor, with flow rates between 20 and 40 L/min. At lower flow rates (20 L/min), no issues with elevated $PtcCO_2$ were observed. However, at higher flow rates (40 L/min), $PtcCO_2$ dropped considerably during and after exercise therapy, and discomfort was reported (Fig 2). Therefore, the flow rate in this study was set at 20 L/min.

## Calculation of the number of subjects needed

Power-based sample size calculations for crossover randomized can be performed using the methods described by Grady et al. [31,49]. Calculating the required sample sizes is essential for feasibility assessments in pilot studies. To determine

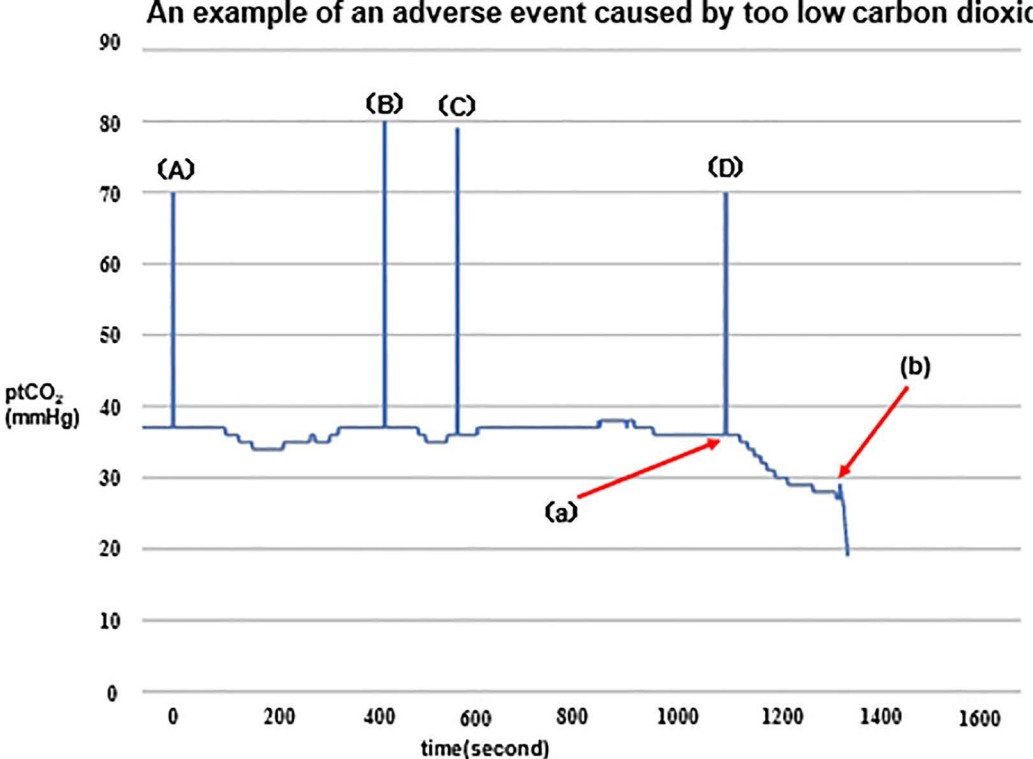

**Fig 2. An example of an adverse event caused by hypocapnia.** (A) $O_2$ loading (Flow rate 40 L/m). Stretching and Resistance training begins. (B) Stretching and resistance training completed. (C) Aerobic exercise begins. (D) Aerobic exercise completed. (a) During aerobic exercise, $CO_2$ levels were low. Seventeen minutes and 30 seconds after the start of the treadmill, the patient complained of dizziness, and the exercise was stopped. (b) $CO_2$ levels decreased further after exercise cessation. The subject continued to report dizziness.

the required sample size, standardized effect sizes were calculated [31,49,50], targeting a significance level of 0.05 and a power of 0.8. We assumed that both methods (Program A and Program B) would be effective and that their effectiveness would differ. Initially, the 6MWD was selected as the primary endpoint; however, as the study progressed, we reconsidered its suitability due to cases where 6MWD showed no improvement after PR. Our aim then shifted to identifying a more appropriate indicator that demonstrated significant gains post-PR to use in calculating the required sample size. A larger standardized effect size corresponds to a smaller sample size, indicating better feasibility [31,49,50]. The change in Program B, including $FiO_2$ 0.5 PR, compared to that in Program A, including $FiO_2$ 0.3 PR, was defined as the effect size (E), and the standard deviation (S) was calculated from all measurements in the six patients with COPD. The standardized effect size was then calculated as E/S.

## Statistical analysis

Data are expressed as mean ± standard deviation unless otherwise indicated. A two-tailed Student's paired t-test was used to compare between baseline and after walking rehabilitation. Additionally, a sub-analysis was conducted to assess correlations among evaluation indices using Spearman's rank correlation coefficient. Statistical significance was set at $p < 0.05$. Commercially available statistical software (BellCurve for Excel; Social Survey Research Information Co., Ltd., Tokyo, Japan) and SPSS version 25 (IBM Corp., Armonk, NY, USA) were used for the statistical analyses.

## Results

### Profiles of participants

The Global Initiative for Chronic Obstructive Lung Disease (GOLD) grades of the six participants (4 male, 2 female; mean age 76.0 years, 95% confidence interval [71.8–80.2]) in the study were as follows: GOLD 1 (Mild, FEV1 ≥ 80% predicted), 3 cases (2 male, 1 female); GOLD 2 (Moderate, 50% ≤ FEV1 < 80% predicted), 1 case (female); GOLD 3 (Severe, 30% ≤ FEV1 < 50% predicted), 2 cases (1 male, 1 female); GOLD 4 (Very severe, FEV1 < 30% predicted), 0 cases. The other profiles of the participants are presented in Table 1. Three patients initially underwent the $FiO_2$ 0.3 PR, and following the washout and crossover, they underwent the $FiO_2$ 0.5 PR (Fig 1). The other three patients initially underwent the $FiO_2$ 0.5 PR, and after the washout and crossover, they underwent the $FiO_2$ 0.3 PR (Fig 1).

### Possible primary endpoints before and after PR intervention

Changes in various parameters before and after the PR intervention are shown in S1 File. The 6MWD, CAT, and items showing significant improvement before and after the intervention were evaluated for possible primary endpoints.

The results of the 6MWD and CAT before and after Programs A and B, including PR under $O_2$ supplementation with HFNC, are shown in Fig 3. No significant difference in 6MWD was observed between pre- and post-intervention for Programs A and B (Fig 3A). The change in 6MWD from pre- to post-intervention (Δ6MWD) for Program B, including $FiO_2$ 0.5 PR, was −6.5 ± 59.9 m, indicating a decrease from baseline performance (Fig 3B). The CAT score markedly decreased following the intervention, from 16.9 ± 7.9 pre-intervention to 10.0 ± 5.9 post-intervention (Fig 3C). ΔCAT was −8.2 ± 6.5 for Program A, including $FiO_2$ 0.3 PR, and −5.7 ± 9.4 for Program B, including $FiO_2$ 0.5 PR, indicating greater improvement with the Program A than with the Program B (Fig 3D). Lower scores indicate better CAT scores.

**Table 1. Patient characteristics.**

| First assigned $FiO_2$ | 0.3 (n = 3) | 0.5 (n = 3) |
|---|---|---|
| Age (years) | 74.7 ± 5.0 | 77.3 ± 6.1 |
| Male | 2 | 1 |
| BMI | 21.4 ± 0.2 | 22.7 ± 2.2 |
| Smoking (pack years) | 36.8 ± 23.0 | 80.0 ± 17.8 |
| FEV1/FVC (%) | 64.3 ± 4.0 | 47.7 ± 12.3 |
| 6MWD (m) | 334 ± 80 | 272 ± 50 |
| CAT score | 17.8 ± 9.8 | 18.7 ± 1.5 |
| Comorbidity | | |
| Hypertension | 3 | 2 |
| Dyslipidemia | 3 | 2 |
| Diabetes mellitus | 0 | 0 |
| Arterial blood gas analysis | | |
| $PaO_2$ (mmHg) | 73.4 ± 5.0 | 64.1 ± 10.0 |
| $PaCO_2$ (mmHg) | 33.8 ± 3.8 | 44.8 ± 4.9 |
| pH | 7.45 ± 0.02 | 7.42 ± 0.02 |

Some data are expressed as mean ± standard deviation.

Abbreviations: BMI, body mass index; CAT, chronic obstructive pulmonary disease assessment test; FEV1, forced expiratory volume in 1 s; FVC, forced vital capacity; HFNC, high-flow nasal cannula; $PaCO_2$, partial pressure of arterial carbon dioxide; $PaO_2$, partial pressure of arterial oxygen; PR, pulmonary rehabilitation; 6MWD, 6-min walking distance.

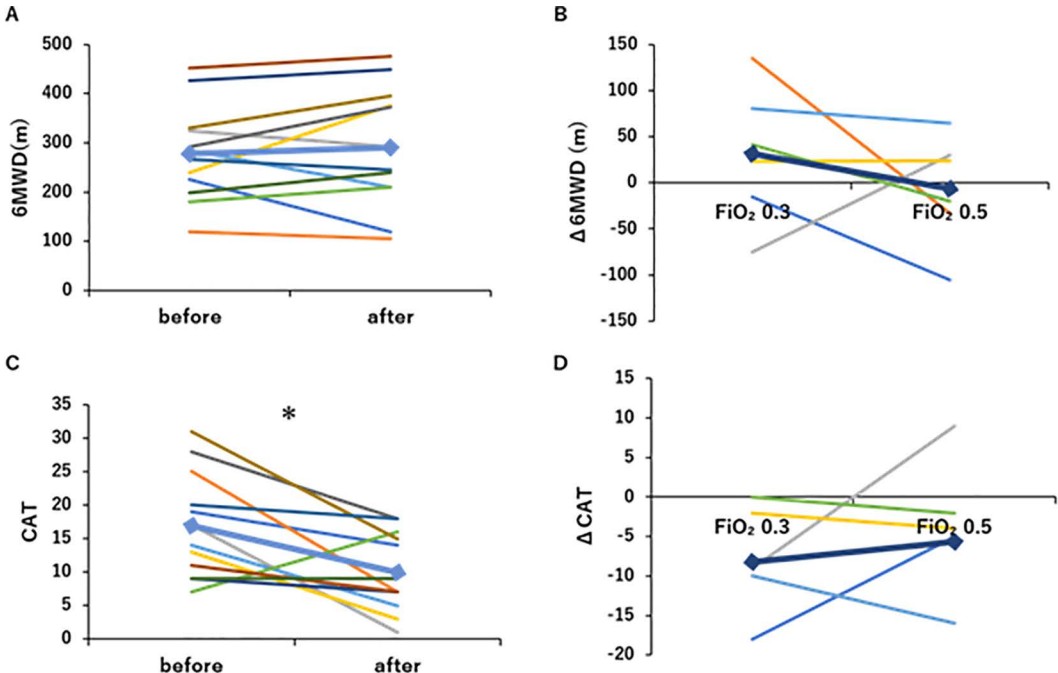

**Fig 3. Changes in six-min walk distance (6MWD) and COPD assessment (CAT) score by O$_2$-supplemented pulmonary rehabilitation (PR).** (A) Program A or B, including PR with O$_2$ supplementation (FiO$_2$ 0.3 or 0.5, respectively), did not significantly increase 6MWD (n = 6). (B) The Δ6MWD for Program B, including FiO2 0.5 PR, indicated a decrease from baseline performance (n = 6). (C) Program A or B, including PR with O$_2$ supplementation (FiO$_2$ 0.3 or 0.5, respectively), significantly decreased CAT score (n = 6). (D) The decrease in CAT score after Program A, including FiO$_2$ 0.3 PR, was larger than that after Program B, including FiO$_2$ 0.5 PR (n = 6). *: p < 0.05, Line graph: ♦Mean. Colors represent each subject.

The left and right knee extensor strength results are presented in Fig 4. Right knee extensor strength (rt quadriceps muscle power) significantly increased from 25.0 ± 8.4 kg to 29.9 ± 10.0 kg (p < 0.05, Fig 4A), and Δ (rt quadriceps muscle power) increased in an O$_2$-dependent manner (Fig 4B). Left knee extensor strength (lt quadriceps muscle power) significantly increased from 25.1 ± 10.3 kg to 29.0 ± 11.2 kg after the intervention (p < 0.05, Fig 4C), and Δ (lt quadriceps muscle power) increased in an O$_2$-dependent manner (Fig 4D). Overall, both right and left quadriceps muscle power showed significant improvement after the intervention (p < 0.01, Fig 4E), with Δ (rt or lt quadriceps muscle power) displaying an O$_2$-dependent increase (Fig 4F).

The left and right calf circumference results are shown in Fig 5. The right calf circumference significantly increased from 32.5 ± 2.5 cm to 33.0 ± 2.8 cm after the intervention (p < 0.01, Fig 5A), and Δ (rt calf circumference) increased in an O$_2$-dependent manner (Fig 5B). The left calf circumference significantly increased from 32.5 ± 2.7 cm to 33.1 ± 2.8 cm after the intervention (p < 0.01, Fig 5C), and Δ (lt calf circumference) increased in an O$_2$-dependent manner (Fig 5D). Overall, both right and left calf circumferences showed significant improvement after the intervention (p < 0.001, Fig 5E), with Δ (rt or lt calf circumference) displaying an O$_2$-dependent increase (Fig 5F).

The muscle mass results of the trunk and both lower limbs are presented in Fig 6. The trunk muscle mass significantly increased from 20.2 ± 2.6 kg to 20.4 ± 2.5 kg (p < 0.05, Fig 6A), and Δ (trunk muscle mass) increased in an O$_2$-dependent manner (Fig 6B). The right lower limb muscle mass significantly increased from 5.8 ± 1.2 kg to 5.9 ± 1.2 kg (p < 0.05, Fig 6C), and Δ (rt leg muscle mass) increased in an O$_2$-dependent manner (Fig 6D). Overall, both right and left lower limb muscle masses were significantly increased (p < 0.01, Fig 6E), with Δ (rt or lt leg muscle mass) demonstrating an O$_2$-dependent increase (Fig 6F).

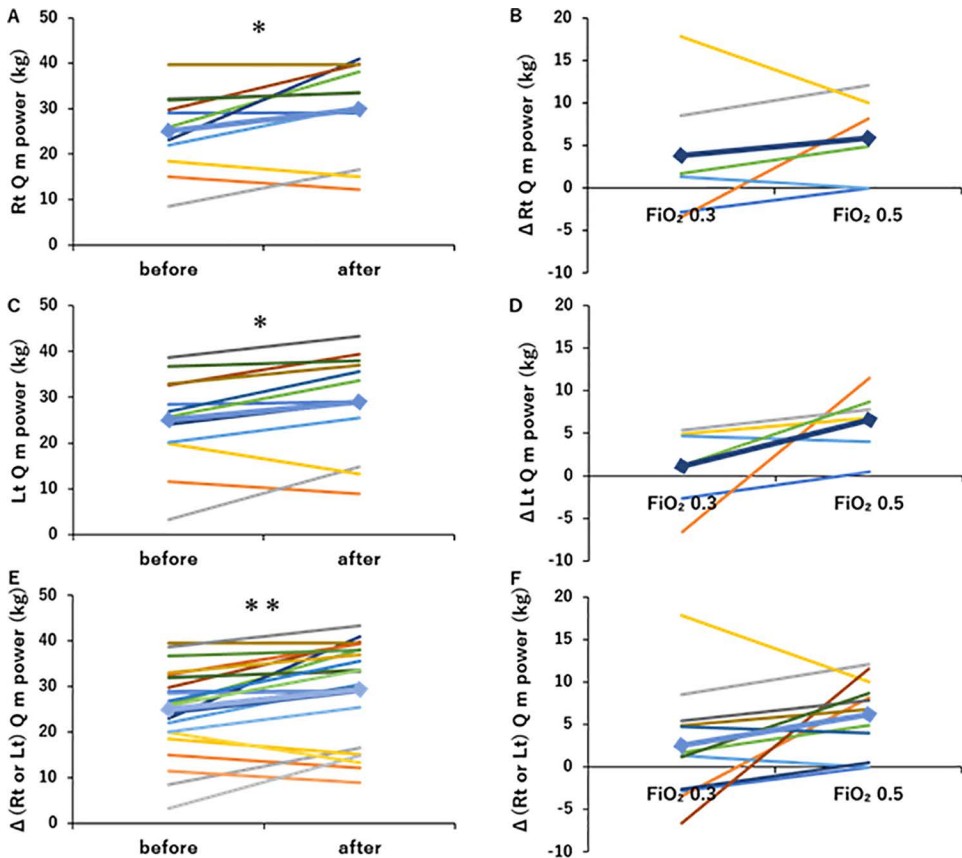

**Fig 4. Changes in quadriceps muscle power by $O_2$-supplemented pulmonary rehabilitation (PR).** (A) Program A or B, including PR with $O_2$ sup-plementation (FiO$_2$ 0.3 or 0.5, respectively), significantly increased rt quadriceps muscle power (n=6). (B) $O_2$-supplemented PR increased rt quadriceps muscle power in a dose-dependent manner (n=6). (C) Program A or B, including PR with $O_2$ supplementation (FiO$_2$ 0.3 or 0.5, respectively), significantly increased lt quadriceps muscle power (n=6). (D) $O_2$-supplemented PR increased lt quadriceps muscle power in a dose dependent manner (n=6). (E) Program A or B, including PR with $O_2$ supplementation (FiO$_2$ 0.3 or 0.5, respectively), significantly increased (rt or lt) quadriceps muscle power (n=12 from 6 patients). (F) $O_2$-supplemented PR increased (rt or lt) quadriceps muscle power in a dose-dependent manner (n=12 from 6 patients). *: p<0.05, **: p<0.01, Line graph: ◆Mean. Colors represent each subject.

No parameters in the lung function tests showed significant changes between pre- and post-$O_2$-supplemented PR inter-ventions (S1 File, latter part).

### Change in physical function before and after PR with $O_2$ supplementation and sample size calculation

The required sample size could not be calculated for the change in 6MWD. For the change in CAT score, the effect size (E/S) was 0.32, with a two-tailed sample size of 81 (Table 2). Because all the muscle-related items in Table 3 significantly increased in an $O_2$-dependent manner with $O_2$ supplementation (S1 File), we adopted the one-tailed calculation for these items. For the change in knee extension strength, the sample size (one-tailed) was 70 with an E/S of 0.30 for the right leg only and 7 with an E/S of 1.07 for the left leg only (one-tailed). For combined left and right leg sampling, an E/S of 0.66 required a sample size of 8 (one-tailed). For the change in lower limb circumference, sampling both legs required a sample size of 8 (one-tailed), with an E/S of 0.66. Sampling only the right leg required 38 participants (one-tailed) with an E/S of 0.29, while sampling only the left leg required 310 (one-tailed) with an E/S of 0.035. For combined left and right leg sampling, an E/S of 0.17 required a sample size of 114 (one-tailed). For muscle mass, trunk muscle mass changes had

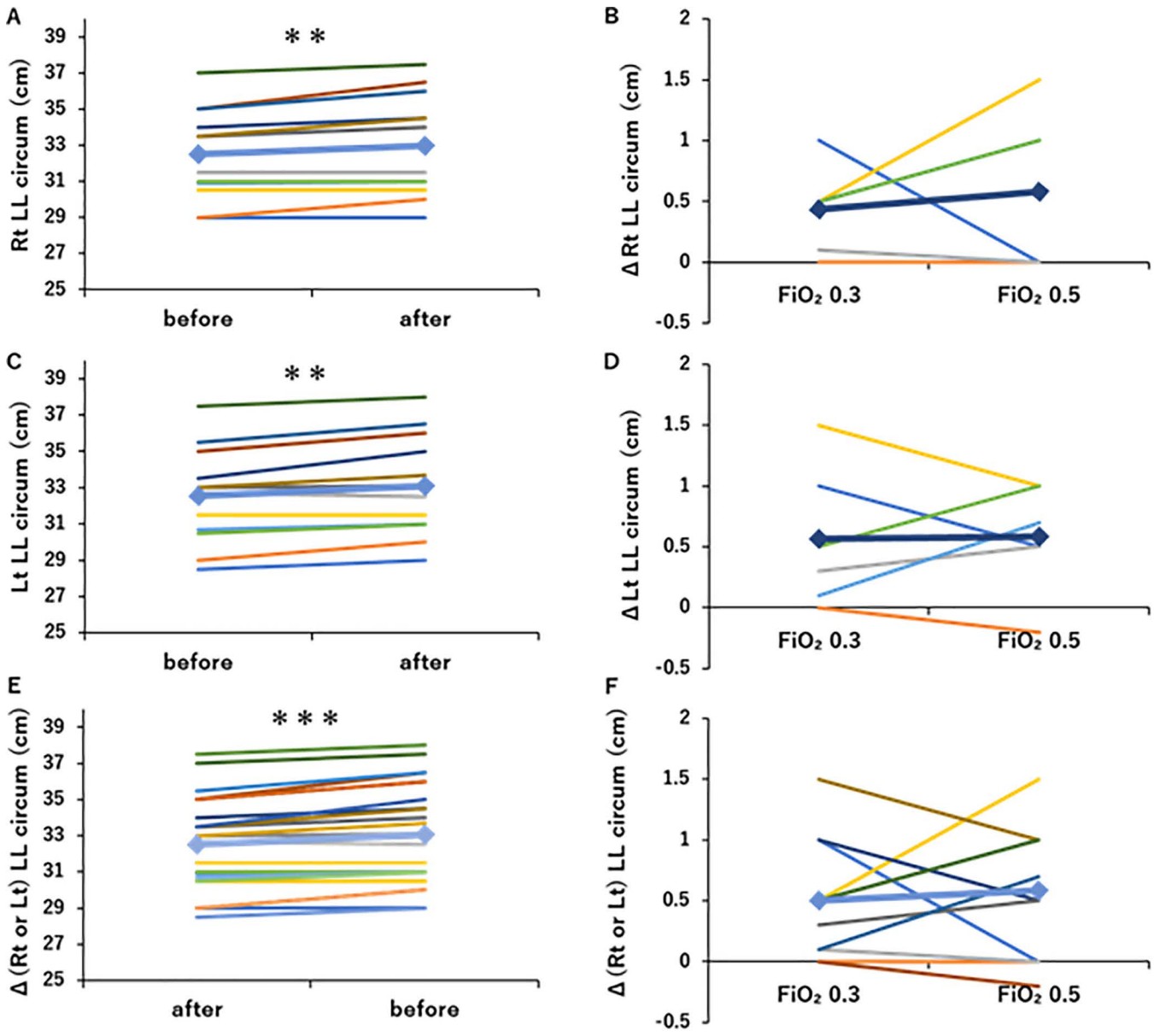

**Fig 5. Changes in lower leg circumference by O$_2$-supplemented pulmonary rehabilitation (PR).** (A) Program A or B, including PR with O$_2$ supplementation (FiO$_2$ 0.3 or 0.5, respectively), significantly increased rt lower leg circumference (n = 6). (B) O$_2$-supplemented PR increased rt lower leg circumference in a dose-dependent manner (n = 6). (C) Program A or B, including PR with O$_2$ supplementation (FiO$_2$ 0.3 or 0.5, respectively), significantly increased lt lower leg circumference (n = 6). (D) O$_2$-supplemented PR increased lt lower leg circumference in a dose-dependent manner (n = 6). (E) Program A or B, including PR with O$_2$ supplementation (FiO$_2$ 0.3 or 0.5, respectively), significantly increased rt and lt lower leg circumference (n = 12 from 6 patients). (F) O$_2$-supplemented PR increased rt and lt lower leg circumference in a dose-dependent manner (n = 12 from 6 patients). **: $p < 0.01$, ***: $p < 0.001$, Line graph: ◆Mean. Colors represent each subject.

an E/S of 0.27 with a sample size (one-tailed) of 88, and the right lower extremity muscle mass had an E/S of 0.24 with a sample size (one-tailed) of 111. Sampling both lower extremities required a sample size of 56 (one-tailed) with an E/S of 0.24 (Table 3). These sample sizes were calculated for a one-tailed test; for a two-tailed test, the required sample size would be double.

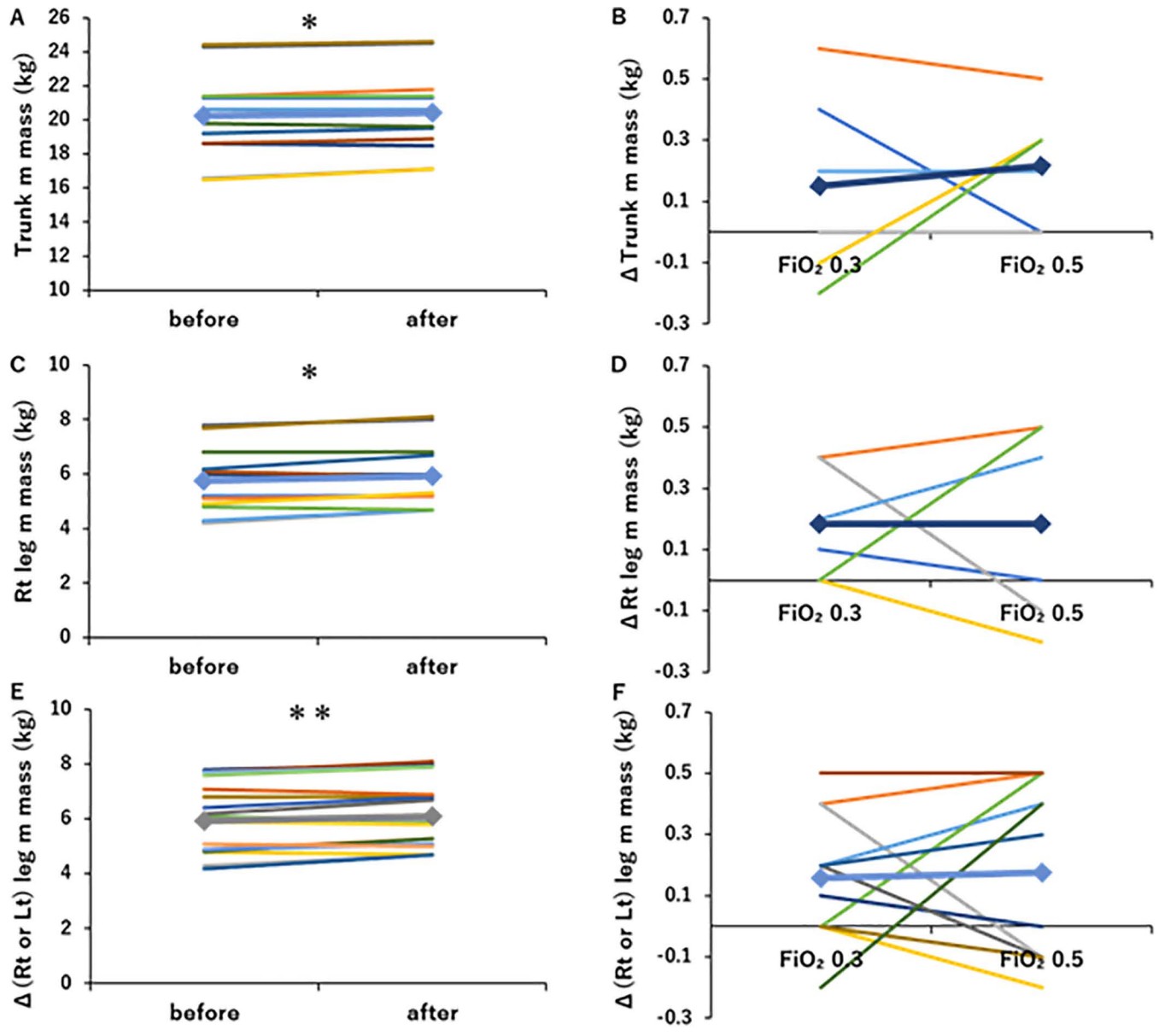

**Fig 6. Changes in muscle mass by O₂-supplemented pulmonary rehabilitation (PR).** (A) Program A or B, including PR with O₂ supplementation (FiO₂ 0.3 or 0.5, respectively), significantly increased trunk muscle mass (n = 6). (B) O₂-supplemented PR increased trunk muscle mass in a dose-dependent manner (n = 6). (C) Program A or B, including PR with O₂ supplementation (FiO₂ 0.3 or 0.5, respectively), significantly increased rt leg muscle mass (n = 6). (D) O₂-supplemented PR increased rt leg muscle mass in a dose-dependent manner (n = 6). (E) Program A or B, including PR with O₂ supplementation (FiO₂ 0.3 or 0.5, respectively), significantly increased rt and lt leg muscle mass (n = 12 from 6 patients). (F) O₂-supplemented PR dose-dependently increased (rt and left) leg muscle mass (n = 12 from 6 patients). *: $p < 0.05$, **: $p < 0.01$, Line graph: ◆Mean. Colors represent each subject.

## Discussion

We initially attempted to use the 6MWD as the primary endpoint to compare the effects of PR with two different oxygen addition doses; however, the 6MWD did not prove suitable for achieving our objectives. Most patients with COPD are older and may experience limitations from factors such as pain from co-existing musculoskeletal or neurological disorders,

**Table 2. Conventional items and the sample size required to demonstrate the differences in rehabilitation achievements between FiO$_2$ 0.5 and FiO$_2$ 0.3 oxygen-supplemented pulmonary rehabilitation (PR) for patients with chronic obstructive pulmonary disease (COPD).**

| Item | Standardized effect size | Number of subjects required |
|---|---|---|
| Six-minute walk distance (6MWD) | N/A. The Δ6MWD for the FiO$_2$ 0.5 PR indicated a decrease from baseline performance. | N/A |
| COPD assessment test score | 0.32. Improvement after FiO$_2$ 0.3 PR was greater than that after FiO$_2$ 0.5 PR. | 81 (two-tailed) |

Confidence level 95% and power 0.8 were sought.

**Table 3. Candidates of new items and the sample size required to demonstrate the differences in rehabilitation achievements between FiO$_2$ 0.5 and FiO$_2$ 0.3 oxygen-supplemented pulmonary rehabilitation (PR) for patients with chronic obstructive pulmonary disease (COPD).**

| Item | Standardized effect size | Number of subjects required |
|---|---|---|
| Rt quadriceps muscle power | 0.30 | 70 (one-tailed) |
| Lt quadriceps muscle power | 1.07 | 7 (one-tailed) |
| (Rt or Lt) quadriceps muscle power | 0.66 | 8[a] (one-tailed) |
| Rt lower limb circumference | 0.29 | 38 (one-tailed) |
| Lt lower limb circumference | 0.035 | 310 or more (one-tailed) |
| (Rt or Lt) lower limb circumference | 0.17 | 114[a] (one-tailed) |
| Trunk muscle mass | 0.27 | 88 (one-tailed) |
| Rt leg muscle mass | 0.24 | 111 (one-tailed) |
| (Rt or Lt) leg muscle mass | 0.24 | 56[a] (one-tailed) |

[a] If we assume that each patient provides separate data for both sides [Rt and Lt]

Confidence level 95% and power 0.8 were sought

as well as breathlessness caused by mild respiratory infections, heart disease, or cancer [51]. These factors make accurate sample size calculation difficult or even impossible. The 6MWD, influenced by many confounding factors, may not accurately reflect the long-term effects of PR. While the CAT is still commonly used to assess PR in COPD, it remains a subjective symptom assessment rather than an objective measure.

Among the items assessed in the present study, those focusing on lower limb muscle strength, which is an objective measure, could serve as direct and accurate indicators of PR achievement. Future PR studies may benefit from using these items as primary endpoints. Between thigh muscle strength and lower leg circumference, thigh muscle strength demonstrated a large effect size in assessing PR achievement and proved to be practical for calculating the required sample size.

It has been reported that quadriceps strength can be improved through conventional exercise therapy [52,53]. Furthermore, a correlation between COPD severity and prognosis has been demonstrated, suggesting that it may be a useful indicator [54].

Both thigh muscle strength and lower leg circumference can be measured in a sitting position, which could be a valid index for patients with COPD with advanced gait difficulties. For patients with COPD, an additional assessment focusing on lower limb muscle strength was considered to allow a more detailed assessment of which rehabilitation plan would be better and to what extent is actually being achieved.

Of the nine items showing significant improvement before and after intervention, the smallest P value was observed for the left and right lower leg circumference. Despite the statistical significance, the difference between the FiO$_2$ 0.3 PR

group and the $FiO_2$ 0.5 PR group was small. The study aimed to compare the two $FiO_2$ groups on various outcomes, and the most pronounced difference was in quadriceps muscle power (standardized effect size), indicating that the level of oxygen supplementation had a larger effect on muscle strength than on other outcomes. We concluded that using thigh muscle strength as the primary endpoint would be the most feasible approach, as it would require fewer participants to achieve statistically meaningful results, thus optimizing the study's efficiency.

Given that several participants required a higher $FiO_2$ for adequate exercise therapy (i.e., higher than $FiO_2$ 0.3), it was reasonable to assume that the improvement in the muscular system index values would be $O_2$ dose-dependent. However, the CAT, the index value for subjective symptoms, did not improve in an $O_2$ amount-dependent manner; $FiO_2$ 0.3 PR showed greater improvement than $FiO_2$ 0.5 PR. To further investigate this phenomenon, we examined the correlation between the ΔCAT and the change in various index values and found that thigh muscle strength exhibited a marked positive correlation, whereas trunk muscle mass demonstrated a marked inverse correlation. CAT was more indicative of improvement with decreasing values, which suggests that trunk muscle mass is an important indicator. However, improving thigh muscle strength could potentially worsen subjective symptoms, as it may increase the physical load during daily activities. Further research with a larger sample size is required to confirm this hypothesis.

We examined the changes in pulmonary function test scores and found no significant differences between pre- and post-PR, and none of these scores was useful as a primary endpoint. Although some previous studies have reported significant before-and-after differences [55], we believe that pulmonary function tests currently do not provide a useful primary endpoint for a rigorous study design.

Regarding the safety of HFNC, none of the participants complained of any discomfort except for dizziness, numbness, and headache caused by hyperventilation during the preliminary study with a flow rate of 20 L/m. We set 20 L/m as the basic flow rate. When $PtcCO_2$ was high, the flow rate might be increased to 40–50 L/m to prevent $CO_2$ accumulation. When $PtcCO_2$ was low, it was considered best to discontinue HFNC immediately once rehabilitation is complete.

Although this was a pilot study, we found that using thigh muscle strength as the primary endpoint would allow for future studies to be conducted without a substantial increase in participant numbers. However, increasing the sample size would enhance the ability to examine the correlations between ΔCAT and changes in various other indices. We are considering expanding the participant pool and preparing a final report.

There is a consensus that the 6MWD correlates with COPD prognosis, whereas the relationship between the lower extremity muscle strength system, trunk muscle mass, and COPD prognosis remains less clear. Some studies have reported improvements in lower extremity muscle strength with respiratory rehabilitation [56,57]. However, existing data on the relationship between various muscle strength parameters and prognosis in patients with COPD may be insufficient, warranting further in-depth evaluation. Muscle strength, particularly thigh muscle strength, may be useful in designing studies to assess more effective methods of respiratory rehabilitation.

As a pilot study, this research has some key limitations. First, the small sample size restricts the ability to draw definitive conclusions and limits the generalizability of the findings. Additionally, the use of the 6MWD and the CAT, both of which are influenced by a range of confounding factors such as comorbidities and subjectivity, may have compromised the accuracy of assessing the true effects of PR. Future studies should address these limitations by increasing the sample size to improve statistical power and the generalizability of the findings.

## Conclusion

By conducting a pilot double-blind, crossover, randomized controlled trial comparing two oxygen-supplemented PR in patients with COPD, we found that quadriceps muscle power seemed useful as a primary endpoint for this study's feasibility. We conclude that researchers should not focus solely on 6MWD and should thoroughly investigate how quadriceps muscle strength is related to the prognosis of COPD.

## Supporting information

**S1 File. Step-by-step protocol (also available on protocols.io).**
(PDF)

**S2 File. Changes before and after oxygen-supplemented pulmonary rehabilitation.** Data on lung function tests are shown in the latter part of this table. *: $p<0.05$, **: $p<0.01$, ***: $p<0.001$. Abbreviations: 6MWD, Six-Minute Walk Distance; CAT, COPD Assessment Test; mMRC, Modified Medical Research Council Dyspnea Scale; BMI, Body Mass Index; VC, Vital Capacity; ERV, Expiratory Reserve Volume; IRV, Inspiratory Reserve Volume; TV, Tidal Volume; IC, Inspiratory Capacity; FVC, Forced Vital Capacity; FEV1.0, Forced Expiratory Volume in 1 Second; PEF, Peak Expiratory Flow; V75, Expiratory Flow at 75% of Forced Vital Capacity; V50, Expiratory Flow at 50% of Forced Vital Capacity; V25, Expiratory Flow at 25% of Forced Vital Capacity; MMF, Maximal Mid-Expiratory Flow.
(DOCX)

**S3 File. Results of Spearman's rank correlation coefficients between each preintervention assessment index.** Abbreviations: 6MWD, Six-Minute Walk Distance; CAT, COPD Assessment Test; mMRC, Modified Medical Research Council Dyspnea Scale; Rt, Right; Lt, Left; BMI, Body Mass Index.
(DOCX)

**S4 File. Spearman's rank correlation coefficient results between each pre-post change measure.** Abbreviations: 6MWD, Six-Minute Walk Distance; CAT, COPD Assessment Test; mMRC, Modified Medical Research Council Dyspnea Scale; Rt, Right; Lt, Left; BMI, Body Mass Index.
(DOCX)

**S5 File. Raw data.**
(XLSX)

## Acknowledgments

The authors thank all the medical staff at the International University of Health and Welfare Shioya Hospital and the patients who participated in this study.

## Author contributions

**Conceptualization:** Akihiro Ito, Akane Morito, Masahiro Ishizaka, Akira Umeda.

**Data curation:** Akihiro Ito, Akane Morito, Yukihiro Ogawa, Yuki Kawai, Yuta Hanawa, Naotaka Onodera, Yoshiaki Endo, Taichi Mochizuki, Akira Umeda.

**Formal analysis:** Akihiro Ito, Akane Morito, Yukihiro Ogawa, Yuki Kawai, Yuta Hanawa, Naotaka Onodera, Yoshiaki Endo, Isato Fukushi, Taichi Mochizuki, Yasuo To, Akira Umeda.

**Funding acquisition:** Akihiro Ito, Akira Umeda.

**Investigation:** Akihiro Ito, Akane Morito, Yukihiro Ogawa, Yuki Kawai, Yuta Hanawa, Naotaka Onodera, Yoshiaki Endo, Taichi Mochizuki, Yasushi Inoue, Akira Umeda.

**Methodology:** Akihiro Ito, Akira Umeda.

**Project administration:** Akihiro Ito, Akira Umeda.

**Resources:** Akira Umeda.

**Software:** Akihiro Ito, Akira Umeda.

**Supervision:** Akihiro Ito, Taichi Mochizuki, Akira Umeda.

**Validation:** Akihiro Ito, Isato Fukushi, Kotaro Takeda, Yasushi Inoue, Yasuo To, Seiichiro Sakao, Kazuyuki Chibana, Hideaki Yamasawa, Satoshi Fuke, Sarah Kesler, David Gozal, Yasumasa Okada, Akira Umeda.

**Visualization:** Akihiro Ito, Isato Fukushi, Akira Umeda.

**Writing – original draft:** Akihiro Ito, Akira Umeda.

**Writing – review & editing:** Akihiro Ito, Akane Morito, Masahiro Ishizaka, Yuki Kawai, Isato Fukushi, Kotaro Takeda, Taichi Mochizuki, Yasushi Inoue, Yasuo To, Seiichiro Sakao, Kazuyuki Chibana, Hideaki Yamasawa, Satoshi Fuke, Sarah Kesler, David Gozal, Yasumasa Okada, Akira Umeda.

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
