## [Decision Letter · Decision Letter 0]

30 Dec 2025

PONE-D-25-47949Feasibility assessment of double-blind, crossover, random controlled trial protocol comparing two oxygen-supplemented pulmonary rehabilitation for patients with chronic obstructive pulmonary disease: A pilot studyPLOS One

Dear Dr. Akihiro Ito,

Thank you for submitting your manuscript to PLOS ONE. After careful consideration, we feel that it has merit but does not fully meet PLOS ONE’s publication criteria as it currently stands. Therefore, we invite you to submit a revised version of the manuscript that addresses the points raised during the review process.

We look forward to receiving your revised manuscript.

Kind regards,

Mehrnaz Kajbafvala, Ph.D

Academic Editor

PLOS One

“AI

24K20446

Grants-in-Aid for Scientific Research（KAKENHI）

https://www.jsps.go.jp/j-grantsinaid/

co-first author, co-corresponding author

3. We note you have not yet provided a protocols.io PDF version of your protocol and/or a protocols.io DOI. When you submit your revision, please provide a PDF version of your protocol as generated by protocols.io (the file will have the protocols.io logo in the upper right corner of the first page) as a Supporting Information file. The filename should be S1_file.pdf, and you should enter “S1 File” into the Description field. Any additional protocols should be numbered S2, S3, and so on. Please also follow the instructions for Supporting Information captions [https://journals.plos.org/plosone/s/supporting-information#loc-captions]. The title in the caption should read: “Step-by-step protocol, also available on protocols.io.”

Please assign your protocol a protocols.io DOI, if you have not already done so, and include the following line in the Materials and Methods section of your manuscript: “The protocol described in this peer-reviewed article is published on protocols.io (https://dx.doi.org/10.17504/protocols.io.[...]) and is included for printing purposes as S1 File.” You should also supply the DOI in the Protocols.io DOI field of the submission form when you submit your revision.

If you have not yet uploaded your protocol to protocols.io, you are invited to use the platform’s protocol entry service [https://www.protocols.io/we-enter-protocols] for doing so, at no charge. Through this service, the team at protocols.io will enter your protocol for you and format it in a way that takes advantage of the platform’s features. When submitting your protocol to the protocol entry service please include the customer code PLOS2022 in the Note field and indicate that your protocol is associated with a PLOS ONE Lab Protocol Submission. You should also include the title and manuscript number of your PLOS ONE submission.

4. We note that there is identifying data in the Supporting Information file < Sample.xlsx>. Due to the inclusion of these potentially identifying data, we have removed this file from your file inventory. Prior to sharing human research participant data, authors should consult with an ethics committee to ensure data are shared in accordance with participant consent and all applicable local laws.

-Location data

Please remove or anonymize all personal, ensure that the data shared are in accordance with participant consent, and re-upload a fully anonymized data set. Please note that spreadsheet columns with personal information must be removed and not hidden as all hidden columns will appear in the published file.

Reviewers' comments:

Reviewer's Responses to Questions

**Comments to the Author**

1. Does the manuscript report a protocol which is of utility to the research community and adds value to the published literature?

Reviewer #1: No

2. Has the protocol been described in sufficient detail?

To answer this question, please click the link to protocols.io in the Materials and Methods section of the manuscript (if a link has been provided) or consult the step-by-step protocol in the Supporting Information files.

The step-by-step protocol should contain sufficient detail for another researcher to be able to reproduce all experiments and analyses.

Reviewer #1: Partly

3. Does the protocol describe a validated method?

Reviewer #1: No

4. If the manuscript contains new data, have the authors made this data fully available?

Reviewer #1: Yes

**5. Is the article presented in an intelligible fashion and written in standard English?**

Reviewer #1: Yes

6. Review Comments to the Author

Reviewer #1: The following comments aim to provide constructive feedback on the scientific content, methodological rigor, and clarity of presentation.

Abstract:

Lines 51–52: One month of regular PR followed by two months of oxygen-supplemented PR does not clearly reflect a crossover design, which typically includes randomized order and washout periods.

Lines 50–52: The inclusion of “physical measurements” is too generic, Abstracts should specify key secondary outcomes, especially in feasibility studies.

Introduction:

Lines 65–67: The statement that COPD has a “global prevalence of more than 10%” is potentially misleading, GOLD reports prevalence varying widely by region, age, and diagnostic criteria.

Lines 68–69: COPD is described as being caused by smoking and air pollution only, Other risk factors (e.g., biomass exposure, occupational dusts) are omitted.

Lines 75–76: Describing PR as strictly a “4- to 12-week supervised exercise program” may be overly narrow, PR includes education, behavioral change, and long-term maintenance strategies.

Lines 94–95: The hypothesis that “both methods would be valid” is methodologically vague, Validity is not a measurable endpoint.

Lines 96–98: Claims about establishing a “new standard of care” are not appropriate for a feasibility pilot study.

Method:

Lines 111–113: “Six participants were consecutively recruited” conflicts with the earlier statement implying 108 patients.

Lines 115–117: “random controlled trial” should be randomized controlled trial.

Lines 118–120: Randomization “by their physicians” raises allocation bias risk.

Only the physical therapist appears to be blinded, this does not constitute double blinding, Participant blinding to FiO₂ levels is not explained.

Lines 126–131: The sequence described does not clearly reflect a balanced crossover design, both groups appear to receive “regular PR” first, Period effects and order randomization are unclear. A washout of “>3 months” is unusually long and not justified with references.

Lines 132–136: A total of six assessments” are claimed, but only four time points are listed.

Lines 161–166: The rationale for including trunk ROM as an outcome is not justified in the context of COPD PR.

Line 172: Once-weekly PR is below standard PR frequency.

Line 204: Too low carbon dioxide” should be replaced with hypocapnia.

Lines 211–222: Power-based sample size calculation is not standard for feasibility studies.

7. PLOS authors have the option to publish the peer review history of their article (what does this mean?). If published, this will include your full peer review and any attached files.

Reviewer #1: No

To ensure your figures meet our technical requirements, please review our figure guidelines: s://journals.plos.org/plosone/s/figures

You may also use PLOS’s free figure tool, NAAS, to help you prepare publication quality figures: s://journals.plos.org/plosone/s/figures#loc-tools-for-figure-preparation.

---

## [Author Response · Author response to Decision Letter 1]

12 Feb 2026

We would like to express our sincere gratitude for reviewing our manuscript. We believe that your useful comments have greatly helped us improve our manuscript. Please provide our point-to-point responses to the reviewers’ comments below. We have further revised the text accordingly. Our responses to the reviewers’ comments are indicated in red font. Corresponding revisions in the text are underlined and highlighted in yellow.

Comments and Suggestions for Authors

We thank you for your careful review of our manuscript and helpful comments. Our responses to your comments are provided below.

1. Does the manuscript report a protocol which is of utility to the research community and adds value to the published literature?

Reviewer #1: No

RESPONSE: Thank you for your comment. To add value to the research on patients with COPD, the text has been revised as follows. Additionally, we have cited relevant references.

Most patients with COPD are older and may experience limitations from factors such as pain from co-existing musculoskeletal or neurological disorders, as well as breathlessness caused by mild respiratory infections, heart disease, or cancer [51]. These factors make accurate sample size calculation difficult or even impossible. (Page 22, Lines 407–410)

51．　Liu X, Cao Y, Shi Y, Ding H. Association between multimorbidity and disability among elderly with chronic obstructive pulmonary disease in Shanghai, China: a cross-sectional study. BMC Geriatr. 2025;25(1): 901. doi: 10.1186/s12877-025-06488-2. (Page 34, Lines 675–678)

2. Has the protocol been described in sufficient detail?

To answer this question, please click the link to protocols.io in the Materials and Methods section of the manuscript (if a link has been provided) or consult the step-by-step protocol in the Supporting Information files.

The step-by-step protocol should contain sufficient detail for another researcher to be able to reproduce all experiments and analyses.

Reviewer #1: Partly

RESPONSE: Thank you for pointing this out. Figure 1 has been amended. Furthermore, the description of the method has been revised in accordance with the subsequent comments.

3. Does the protocol describe a validated method?

Reviewer #1: No

RESPONSE: Further explanation has been provided regarding the oxygen-assisted rehabilitation and the fact that the oxygen-assisted rehabilitation program in this study has been validated, and additional references have been cited accordingly.

Evidence is gradually emerging regarding the intervention effects of PR combined with oxygen therapy [20]. (Page 6, Lines 86–87)

The protocol developed in this study was based on the measurement methods employed in prior research. [29, 30] (Page 8, Lines 131–133)

Abstract

Lines 51–52: One month of regular PR followed by two months of oxygen-supplemented PR does not clearly reflect a crossover design, which typically includes randomized order and washout periods.

RESPONSE: Thank you for pointing this out. The following text has been amended.

In standard rehabilitation, it is necessary to adjust subjective symptoms and exercise load. Therefore, the design was changed to a crossover design incorporating Program A (including PR under FiO₂ 0.3) and Program B (including PR under FiO₂ 0.5).

two PR programs—Program A (including PR under FiO₂ 0.3) and Program B (including PR under FiO₂ 0.5)—using high-flow nasal cannula oxygen therapy in patients with COPD and exertional dyspnea (n=6). (Page 4, Lines 51–54)

The improvement in CAT by Program A was greater than that by Program B. The improvements in muscle parameters by Program B were greater than those by Program A. (Page 4, Lines 59–61)

a PR program with oxygen supplementation at FiO₂ 3 (Program A), and the other began a PR program with oxygen supplementation at FiO₂ 0.5 (Program B) (Page 8, Lines 129–131)

A double-blind, crossover, randomized controlled trial comparing Program A, including 4 weeks of usual PR followed by 8 weeks of PR under FiO2 0.3, with Program B, including 4 weeks of usual PR followed by 8 weeks of PR under FiO2 0.5 (Page 9, Lines135-137)

(Program A and Program B) (Page 13, Line 239. Page 16, Line283, 285.)

Program A (Page 16, Lines 289, 291, 300)

Program B (Page 16, Lines 286, 290, 291, 297. Page17, Lines301.)

Program A or B (Page 16, Lines 296, 298. Page 17, Lines 316, 318, 321. Page 18, Lines 338, 340, 343. Page 19, Lines 358, 360, 363.)

Lines 50–52: The inclusion of “physical measurements” is too generic, Abstracts should specify key secondary outcomes, especially in feasibility studies.

Introduction:

RESPONSE: Thank you for your comment. Accordingly, the text has been revised as follows.

muscle strength, body composition analysis, respiratory function, and joint range of motion (Page 4, Lines 54–55)

Lines 65–67: The statement that COPD has a “global prevalence of more than 10%” is potentially misleading, GOLD reports prevalence varying widely by region, age, and diagnostic criteria.

RESPONSE: Thank you for your comment. The Introduction section and citations in the relevant text have been revised as follows.

Chronic obstructive pulmonary disease (COPD) is among the top three causes of deaths worldwide, alongside cardiovascular diseases and cancer, with nearly 90% of COPD-related deaths occuring in low- and middle-income countries [1]. The prevalence of COPD varies widely by region, age, and the availability of diagnostic spirometry [1]. (Page 5, Lines 70–73)

1.Global Initiative for Chronic Obstructive Lung Disease (GOLD). GLOBAL GLOBAL STRATEGY FOR THE DIAGNOSIS, MANAGEMENT, AND PREVENTION OF CHRONIC OBSTRUCTIVE PULMONARY DISEASE: 2026 Report [Internet]. 2023. Available from: https://goldcopd.org/2026-gold-report-and-pocket-guide/. (Page 26, Lines 497–501)

Lines 68–69: COPD is described as being caused by smoking and air pollution only, Other risk factors (e.g., biomass exposure, occupational dusts) are omitted.

RESPONSE: Thank you for your comment. Accordingly, the text has been revised as follows.

COPD is caused by smoking, air pollution, biomass exposure, occupational dusts, and host factors (including abnormal lung development and lung aging) and is characterized by reduced alveolar ventilation due to alveolar destruction and ventilation–blood flow imbalance [1, 2]. (Page 6, Lines 75–78)

Lines 75–76: Describing PR as strictly a “4- to 12-week supervised exercise program” may be overly narrow, PR includes education, behavioral change, and long-term maintenance strategies.

RESPONSE: We thank the reviewer for this important comment. We agree that pulmonary rehabilitation is a comprehensive intervention that includes education, behavioral change, and long-term maintenance strategies, in addition to exercise training. In this manuscript, we intended to describe the supervised exercise training component of PR commonly implemented in clinical trials involving patients with stable COPD. To clarify this point, we have revised the text accordingly.

The PR program for patients with stable COPD typically comprises supervised exercise training lasting approximately 4–12 weeks, generally combining strength training with aerobic exercise [13-15]. (Page 5, Lines 83–85)

Lines 94–95: The hypothesis that “both methods would be valid” is methodologically vague, Validity is not a measurable endpoint.

RESPONSE: Thank you for your comment. Accordingly, the text has been revised as follows.

both methods would be effective（Page 7, Line 104）

Lines 96–98: Claims about establishing a “new standard of care” are not appropriate for a feasibility pilot study.

RESPONSE: Thank you for pointing this out. The relevant section has been deleted.

Method:

Lines 111–113: “Six participants were consecutively recruited” conflicts with the earlier statement implying 108 patients.

RESPONSE: Thank you for your comment. No mention of 108 patients was found within this paper. Could you please clarify the reference so that we can make any necessary adjustments?

Lines 115–117: “random controlled trial” should be randomized controlled trial.

RESPONSE: Thank you for your comment. Accordingly, the text has been revised as follows.

2.2 Protocol of a double-blind, crossover, randomized controlled trial comparing two oxygen-supplemented PR programs (Page 8, Lines 126–127)

Feasibility assessment of double-blind, crossover, randomized controlled trial protocol comparing two oxygen-supplemented pulmonary rehabilitation for patients with chronic obstructive pulmonary disease: A pilot study (Page 1, Lines 1–5)

Lines 118–120: Randomization “by their physicians” raises allocation bias risk.

Only the physical therapist appears to be blinded, this does not constitute double blinding, Participant blinding to FiO₂ levels is not explained.

RESPONSE: Thank you for pointing this out. Accordingly, the text has been revised as follows.

Study participants were randomly assigned to one of the two groups using random number tables:… (Page 8, Lines 128–129)

These assignments were concealed from both the participants and the physical therapists responsible for the PR program. (Page 8, Lines 132–133)

Lines 126–131: The sequence described does not clearly reflect a balanced crossover design, both groups appear to receive “regular PR” first, Period effects and order randomization are unclear. A washout of “>3 months” is unusually long and not justified with references.

RESPONSE: Thank you for your comment. In crossover trials, a washout period must be established to ensure the carryover effect from the first administration is substantially eliminated, and this period must be appropriately set according to the nature of the intervention [31]. Considering studies indicating that the learning effects of exercise therapy diminish after a period equivalent to the intervention duration [32] and the duration over which aerobic exercise effects fade [33], the interval was set at three months.

Based on these considerations, the main text has been revised as follows.

The washout period was set according to prior research to allow sufficient time for the learning effects of exercise therapy and the intervention effects of aerobic exercise to dissipate [31-33]. (Page 9, Lines 148–150)

Lines 132–136: A total of six assessments” are claimed, but only four time points are listed.

RESPONSE: Thank you for pointing this out. Accordingly, the text has been revised as follows.

A total of six assessments were conducted: at baseline; after the first course of standard rehabilitation; after the first course of O2-supplemented PR; after the washout period; after the second course of standard rehabilitation; and at the end of the second course of O2-supplemented PR. (Page 9, Lines 151–154)

Additionally, as respiratory function was measured only four times, we have added a separate paragraph.

Respiratory function was assessed on four occasions: at baseline, after the first course of PR, after the washout period, and at the end of the second course of PR. (Page 9, Lines 156–158)

Each measured data point is recorded in the published file.

Lines 161–166: The rationale for including trunk ROM as an outcome is not justified in the context of COPD PR.

RESPONSE: Thank you for your comment. Previous studies have demonstrated that improvements in chest wall mobility enhance respiratory function and correlate with exercise tolerance (➀Effect of chest wall mobilisation on respiratory muscle function in patients with severe chronic obstructive pulmonary disease (COPD): A randomised controlled trial, ➁Effects of aerobic training combined with respiratory muscle stretching on the functional exercise capacity and thoracoabdominal kinematics in patients with COPD: a randomised and controlled trial). As our study also involves conditioning, we have included changes in exercise tolerance as a measurement item, considering the potential impact of altered thoracic mobility.

Line 172: Once-weekly PR is below standard PR frequency.

RESPONSE: Thank you for your comment. GOLD Report p.75, Line 6, indeed states twice weekly or more, but due to constraints within Japan's healthcare system, the intervention was set at once weekly. Previous studies have demonstrated no difference between twice weekly and once weekly and that rehabilitation intervention is effective even at once weekly; we therefore consider comparative evaluation feasible. Additional references have been added to the main text.

PR was conducted for approximately 40 min once a week [43, 44] (Page 11, Lines 193–194)

43．　Baumann HJ, Kluge S, Rummel K, Klose H, Hennigs JK, Schmoller T et al. Low intensity, long-term outpatient rehabilitation in COPD: a randomised controlled trial. Respir Res. 2012;13(1): 86. doi: 10.1186/1465-9921-13-86. (Page 33, Lines 646–648)

44.　O’Neill B, McKevitt A, Rafferty S, Bradley JM, Johnston D, Bradbury I et al. A comparison of twice- versus once-weekly supervision during pulmonary rehabilitation in chronic obstructive pulmonary disease. Arch Phys Med Rehabil. 2007;88(2): 167-172. doi: 10.1016/j.apmr.2006.11.007. (Page 33, Lines 649–652)

Line 204: Too low carbon dioxide” should be replaced with hypocapnia.

RESPONSE: Thank you for pointing this out. Accordingly, the text has been revised as follows.

Fig 2. An example of an adverse event caused by hypocapnia. (Page 12, Line 226)

Lines 211–222: Power-based sample size calculation is not standard for feasibility studies.

RESPONSE: Thank you for your comment. We have added the references used for the sample size calculation and made the following additions and amendments to the text.

Power-based sample size calculations for crossover randomized can be performed using the methods described by Grady et al. [31, 49]. Calculating the required sample sizes is essential for feasibility assessments in pilot studies. (Page 13, Lines 235–237)

We assumed that both methods (Program A and Program B) would be effective and that their effectiveness would differ. (Page 13, Lines 239–240)

49. Browner WS, Newman TB, Hulley SB. Estimating sample size and power: applications and examples. In: Hulley SB, Cummings SR, Browner WS, Grady DG, Newman TB, editors. Designing clinical research. 4th ed. LIPPINCOTT WILLIAMS & WILKINS, a WOLTERS KLUWER business; 2013. p. 55-83. Two Commerce Square, 2001 Market Street, Philadelphia, PA 19103 USA. (Page 34, Lines 668–671)

Ultimately, we would like to express our sincere gratitude to the editors and reviewers for their positive and constructive criticism. The manuscript has vastly benefited from your valuable and insightful comments and suggestions. We look forward to hearing from you and would be happy to address any further concerns, if required. We hope this further pushes the manuscript closer to publication in your esteemed journal.

---

## [Decision Letter · Decision Letter 1]

15 Apr 2026

Feasibility assessment of double-blind, crossover, randomized controlled trial protocol comparing two oxygen-supplemented pulmonary rehabilitation for patients with chronic obstructive pulmonary disease: A pilot study

PONE-D-25-47949R1

Dear Dr. Ito,

We’re pleased to inform you that your manuscript has been judged scientifically suitable for publication and will be formally accepted for publication once it meets all outstanding technical requirements.

Kind regards,

Jamal Akhtar

Academic Editor

PLOS One

Additional Editor Comments (optional):

Reviewers' comments:

Reviewer's Responses to Questions

**Comments to the Author**

1. Does the manuscript report a protocol which is of utility to the research community and adds value to the published literature?

Reviewer #1: Yes

2. Has the protocol been described in sufficient detail?

To answer this question, please click the link to protocols.io in the Materials and Methods section of the manuscript (if a link has been provided) or consult the step-by-step protocol in the Supporting Information files.

The step-by-step protocol should contain sufficient detail for another researcher to be able to reproduce all experiments and analyses.

Reviewer #1: Yes

3. Does the protocol describe a validated method?

Reviewer #1: Yes

4. If the manuscript contains new data, have the authors made this data fully available?

Reviewer #1: Yes

**5. Is the article presented in an intelligible fashion and written in standard English?**

Reviewer #1: Yes

6. Review Comments to the Author

Reviewer #1: All the questions in the previous version of the manuscript have been addressed by the authors and so it seems now the manuscript is realy to be accepted.

7. PLOS authors have the option to publish the peer review history of their article (what does this mean?). If published, this will include your full peer review and any attached files.

Reviewer #1: No

---

## [Editor Report · Acceptance letter]

PONE-D-25-47949R1

PLOS One

Dear Dr. Ito,

I'm pleased to inform you that your manuscript has been deemed suitable for publication in PLOS One. Congratulations! Your manuscript is now being handed over to our production team.

Kind regards,

on behalf of

Dr. Jamal Akhtar

Academic Editor

PLOS One